# The prevalence and associated factors of the minimum acceptable diet among children aged 6–23 months in Ethiopia: A community-based cross-sectional study

**Girma Cheru Bikila[1], Godana Arero[1], Sultan Kalu[1], Kedir Teji Roba[2], Tesfaye Getachew Charkos[1]***

1 Department of Public Health, Adama Hospital Medical College, Adama, Ethiopia, 2 School of Nursing and Midwifery, College of Health and Medical Sciences, Haramaya University, Harar, Ethiopia

* tesfayegch@gmail.com

## Abstract

### Background

A minimum acceptable diet for children aged 6–23 months is limited globally, with Ethiopia's proportion reducing to one in nine. This study was aimed to assess the prevalence of the minimum acceptable diet and associated factors among children aged 6–23 months in Dera town, Oromia, Ethiopia.

### Methods

A community-based cross-sectional study was conducted. Systematic random sampling techniques were used to select the study subjects. The data was coded, entered into Epi-Info version 7, and then exported to SPSS version 24 for analysis. The variance inflation factor and tolerance test are used to check multicollinearity. Descriptive statistics of frequency (%) were used. Hosmer and Lemeshow's goodness-of-fit test at a P-value > 0.05 is considered the model fit. Bivariate and multivariate logistic regression analyses were computed with a 95% confidence interval, and a P-value < 0.05 was, considered statistically significant.

### Results

A total of 430 study subjects were included in this study. The prevalence of a minimum acceptable diet was 36.5% (95% confidence interval (CI): 32, 41%). In the adjusted model, mothers attaining a primary school (adjusted odds ratio (AOR) = 2.7, 95%CI: 1.3, 4.8), college and above education (AOR = 4.3, 95%CI: 1.4, 13), child age between 12–17 months (AOR = 6.2, 95%CI: 2.80, 13.50) and 18–23 months (AOR = 4.61, 95%CI: 2.04, 10.40), ANC visit four or more (AOR = 2.0, 95%CI: 1.2, 3.4), and not feeding breast (AOR = 0.15, 95% CI: 0.07, 0.31) were significantly associated with meeting the minimum acceptable diet.

**Data Availability Statement:** The minimal data set used to reach the conclusions in our manuscript can be found in the following repository: https://

figshare.com/articles/dataset/Minimum_
acceptable_diet_among_6_to_23_month_child/
25196963?file=44491724.

**Funding:** The author(s) received no specific funding for this work.

**Competing interests:** The authors have declared that no competing interests exist.

## Conclusion

This study showed that the practice of a minimum acceptable diet was low, according to the World Food Program target. Mothers' educational status, antenatal care visits, age of the child, and breastfeeding were the predictors of the minimum acceptable diet.

## Introduction

The minimum acceptable diet (MAD) is a combination of the minimum dietary diversity (MDD) and minimum meal frequency (MMF) for the child aged 6–23 months during the previous day [1]. Minimum dietary diversity assesses food intake among children aged 6–23 months from at least five out of eight food groups during the previous day [2]. Minimum meal frequency is the percentage of children 6–23 months of age who consumed solid, semi-solid, or soft foods at least the minimum number of times during the previous day [1, 2].

Globally, only two-thirds of 6-8-month-olds are receiving any solid food at all; of this, 1 in 2 children receives a minimum meal frequency, and less than 1 in 3 receives minimum dietary diversity. Considering both minimum meal frequency and minimum diet diversity, only about 1 in 6 children receives a minimally acceptable diet [3, 4]. In developing countries, 1 in 5 children aged 6 to 23 months is fed the minimum recommended diverse diet [5]. According to the 2019 Ethiopia mini demographic and health survey (EMDHS), 11% of children aged 6–23 months meet the minimum standards concerning all three infant and young child feeding (IYCF) practices (breastfeeding status, number of food groups, and times) they fed during the day or night before the survey. This indicates only 1 in 9 children receives a minimally acceptable diet, which is very low compared to the national recommendation [6].

At least one in three children is not getting the nutrients they need to grow well, particularly in the crucial first 1,000 days and often beyond [5]. For a child, the first two years are the period during which linear faltering of growth is most prevalent and the period when the process of becoming stunted is almost complete [7]. Inadequate dietary diversity and meal frequency lead children to undernutrition, morbidity, and mortality [6], impaired cognitive development, and low economic productivity in long life [8, 9]. These patterns reflect a profound triple burden of malnutrition that threatens the survival, growth and development of children and of nations [5]. Malnutrition has been responsible, directly or indirectly, for 60% of the deaths annually among children under five. Over two-thirds of these deaths, which were often associated with inappropriate feeding practices, occur during the first year of life [10].

In Ethiopia, more than 50% of deaths in under-five children were caused by inappropriate complementary feeding practices [11]. Undernutrition is also a major cause of disability, preventing children who survive from reaching their full developmental potential [12]. The Ethiopian government set targets to improve the nutritional status of children and to end child malnutrition by 2030 [11]. Despite the above efforts, currently, 37%, 21%, and 7% of under-five-year-old children in Ethiopia are stunted, underweight, and wasted, respectively, which might be due to a low minimum acceptable diet where only 11% of children aged 6–23 months have met the MAD [6].

In developing countries like Ethiopia, high levels of poverty contribute to persistent food insecurity, leading to various health problems. Inadequate feeding practices during infancy and childhood can result in delayed growth and development, stunting, wasting, and undernutrition, which, in turn, increase morbidity and mortality among children under five. Therefore, it is crucial to assess the prevalence and contributing factors of MAD among children aged

6–23 months to mitigate the adverse outcomes associated with inadequate nutritional practices.

## Methods

### Study design and setting

A community-based cross-sectional study was conducted from March 13 to April 30, 2023, in Dera, Oromia, Ethiopia. Dera is located 125 kilometers from Addis Ababa. The district is located in the Great Rift Valley and has an altitude of 1400–2500 meters above sea level with a climatic condition of 99% desert and 1% temperate. Mostly cultivated grains are wheat, teff, maize, and barley, and mostly cultivated vegetables are onions and garlic. Dera has 33714 people (17193 were females and 16521 were males). It has a total of 5539 under-five children; among these, 1804 were children aged 6–23 months.

### Inclusion and exclusion criteria

All children aged 6–23 months with a mother or caregiver living for at least 6 months or more were included in the study. Children whose mothers or caregivers were seriously ill and not able to respond to the interviews were excluded from the study.

### Sample size and sampling technique

The sample size was determined using a single population proportion formula by assuming the proportion of the minimum acceptable diet of 13.8% taken from Aleta wondo District of Sidama Region [13], a confidence interval (CI) of 95%, and a margin of error of 3%. After adding a 10% non-response rate, the final sample size was 436. A systematic sampling technique was used to select the study participants.

### Study variables and measurements

The dependent variable is the minimum acceptable diet. Independent variables include socio-demographic factors (age of the mother, educational status of the mother and the father, monthly income of the household, occupational status of the mother and the father, family size, marital status of the mother, age, and sex of the child). Health service-related factors (ANC visit, PNC visit, GMP follow-up, and health facility delivery) and behavioral factors (media exposure, access to information on IYCF).

### Data collection procedures

Dera town comprises two kebeles, both of which were selected purposively for this study. Study participants were chosen using systematic random sampling. The number of children aged 6 to 23 months with mothers/caregivers during the study period was obtained from the Dera Health Center EPI registration. The sample size was proportionally allocated based on the number of eligible children in each kebele.

   The first respondent was randomly selected using the lottery method. Subsequent respondents were selected every Kth interval, where K = N/n = 1804/436 = 4 (N being the total population in the study area and n being the sample size). From each household, one eligible child with a mother/caregiver was selected. If more than one eligible child was present, one was randomly chosen by lottery method. This process was repeated until the required sample size for both kebeles was reached. If a mother/caregiver was not available on the date of data collection, the next mother was approached after two subsequent visits (S1 Fig).

## Data quality control

The questionnaire was created after reviewing different pieces of literature and then adapted to the local context [2, 14–17]. Data was collected using structured questionnaires. The twenty-four-hour recall method and food frequency questionnaire were used to assess dietary diversity and meal frequency. The questionnaire was translated into the local language (the participants' language) Afan Oromo by a language expert and translated into English to validate its consistency. A pretest on 5% of the sample size was taken in Awash Bishola kebele. Six female nurses as data collectors and two male public health officers as supervisors were involved in the survey after being trained for one day.

## Data processing and analysis

Descriptive statistics like frequency and percentage were used for qualitative variables. A bivariate and multiple binary logistic regression analysis is used to determine the association between independent and dependent variables. The crude odd ratio (COR) and adjusted odd ratio (AOR) were calculated. To determine the factors statistically significantly associated with MAD, the odd ratio at 95% CI was determined using multivariable logistic regression analysis. The goodness of fit for the final model was evaluated using the Hosmer and Lemeshow test, resulting in a p-value of 0.635. This indicates that the model is a good fit for the logistic regression. Multicollinearity was assessed using the Variance Inflation Factor (VIF). All tested variables had VIF values less than 10, indicating that there was no multicollinearity. An adjusted odd ratio with a p-value $< 0.05$ was used to report the significant factors associated with MAD. All analysis was performed using the Statistical Package for Social Science (SPSS version 25).

## Operational definitions

**Minimum dietary diversity.** Percentage of children 6–23 months of age who consumed foods and beverages from at least five out of eight defined food groups during the previous day. Consumption of any amount of food or beverage from a food group is sufficient to "count", i.e. there is no minimum quantity [2, 18, 19].

The eight food groups used for the tabulation of this indicator are [2]

1. Breast milk, 2. Grains; roots; tubers and plantains, 3. Pulses (beans, peas, lentils); nuts and seeds, 4. Dairy products (milk, infant formula, yogurt, cheese), 5. Flesh foods (meat, fish, poultry, organ meats), 6. Eggs, 7. Vitamin-A-rich fruits and vegetables and 8. Other fruits and vegetables.

**Minimum meal frequency.** Percentage of children 6–23 months of age who consumed solid, semi-solid, or soft foods (but also including milk feeds for non-breastfed children) at least the minimum number of times during the previous day [2, 6, 19].

The minimum number of times is defined as:

Two feedings of solid, semi-solid, or soft foods for breastfeeding infants aged 6–8 months and three feedings of solid, semi-solid, or soft foods for breastfeeding children aged 9–23 months. Four feedings of solid, semi-solid, or soft foods or milk feeds for non-breastfeeding children aged 6–23 months whereby at least one of the four feeds must be a solid, semi-solid, or soft feed [2, 18, 19].

**A minimum acceptable diet.** Is a combination of the minimum dietary diversity (MDD) and the minimum meal frequency (MMF) [1, 2]. For breastfed children: receiving at least the minimum dietary diversity and minimum meal frequency for their age during the previous day; for non-breastfed children: receiving at least the minimum dietary diversity and

minimum meal frequency for their age during the previous day as well as at least two milk feeds [2].

**Minimum milk feeding frequency for non-breastfed children 6–23 months (MMFF).** Percentage of non-breastfed children 6–23 months of age who consumed at least two milk feeds⁻ during the previous day.

**Infant and young child feeding practices.** Appropriate IYCFP **includes** early initiation (within one hour of birth) of exclusive breastfeeding, exclusive breastfeeding for the first six months of life, followed by nutritionally adequate and safe complementary foods, while breast-feeding continues for up to two years of age or beyond [1, 20, 21].

**Exposure to communication media.** Communication media encompasses a wide range of channels and tools used to convey information related to child feeding practices, such as digital media, social media, print media, broadcast media, and mobile media. In our study, if mothers had exposure to at least one of these media, they were categorized as "Yes"; otherwise, they were labeled as "No."

## Ethics approval and consent to participate

An ethical issue was obtained from the Institutional Review Board Ethical Committee of Adama Hospital Medical College. Written informed consent was obtained from the parent/guardian of each participant under 18 years of age.

## Results

### Sociodemographic characteristics of the study participants

Four hundred thirty (430) mothers or caregivers with their respective children aged 6–23 months participated in this study (with a response rate of 98.6%). The median age of the mothers or caregivers was 27 years, and 417 (97%) of them lived with their husbands (married). One hundred seventy (39.5%) of the mothers had an elementary level. Three hundred seventy-five (87.2%) of the mothers or caregivers were Oromo and 298 (69.3%) of them were followers of the Muslims. Around 328 (76.3%) of the mothers were housewives. Two hundred fifty-two (58.6%) of the households had less than five family members, and 153 (35.6%) had a monthly income of more than 4501 birr. More than half (50.2%) of the children were female. The majority (35.6%) of the children were in the age group of 12 and 17 months, and the median age of the children was 15 ± 10 months (Table 1).

### Maternal and child healthcare-related characteristics

Four hundred thirty (100%) of the mothers had a history of antenatal care (ANC) follow-up, and 300 (70%) of the mothers had PNC follow-up. Regarding GMP, 404 (94%) of the children had GMP follow-up, and 401 (93.3%) of the children were delivered at health facilities. Of the total participants, 383 (89%), mothers had exposure to communication media, while 43.3% had awareness about infant and young child feeding practice (IYCFP) (Table 2).

### Feeding practice of the children

The majority, 404 (94%) of the parents feed their children grains, roots, and tubers followed by breast milk 339 (78.8%). Only 26 (6%) of the children fed flesh food (organ meat, processed meat, any other meat, and fish or shellfish) was the least consumed food group, which was only consumed by twenty-six (6%) of children (Fig 1).

**Table 1. Socio-demographic characteristics of children aged 6–23 months with a parent in Dera town, Oromia, Ethiopia, March 13 to April 30, 2023. (n = 430).**

| Variables | Category | Frequency (n) | Percent (%) |
|---|---|---|---|
| Mother age in years | ≤ 24 | 114 | 26.5 |
|  | 25–34 | 264 | 61.4 |
|  | ≥ 35 | 52 | 12.1 |
| Marital status of the mother | Married | 417 | 97 |
|  | Single | 13 | 3 |
| Ethnicity of the mother/caregiver | Oromo | 375 | 87.2 |
|  | Silte | 25 | 5.8 |
|  | Amhara | 30 | 7 |
| Religion of the mother/caregiver | Muslim | 298 | 69.3 |
|  | Orthodox | 120 | 27.9 |
|  | Protestant | 12 | 2.8 |
| Educational status of the mother/caregiver | Illiterate | 78 | 18.1 |
|  | Elementary | 170 | 39.5 |
|  | Secondary | 115 | 26.7 |
|  | College and above | 67 | 15.6 |
| Occupation of the mother/caregiver | Housewife | 328 | 76.3 |
|  | Merchant | 60 | 14 |
|  | Government employ | 42 | 9.8 |
| Household member | < 5 | 252 | 58.6 |
|  | ≥ 5 | 178 | 41.4 |
| Father educational status | Illiterate | 43 | 10 |
|  | Elementary | 119 | 27.7 |
|  | Secondary | 142 | 33 |
|  | College and above | 126 | 29.3 |
| Father occupational | Daily laborer | 110 | 25.6 |
|  | Farmer | 54 | 12.6 |
|  | Private | 72 | 16.7 |
|  | Merchant | 110 | 25.6 |
|  | Government employ | 84 | 19.5 |
|  | Other | 18 | 4.1 |
| Household monthly income | ≤ 1500 | 24 | 5.6 |
|  | 1501–3000 | 87 | 20.2 |
|  | 3001–4500 | 61 | 14.2 |
|  | ≥ 4501 | 258 | 60 |
| Sex of the child | Female | 216 | 50.2 |
|  | Male | 214 | 49.8 |
| Age of the child (in months) | 6–8 | 67 | 15.6 |
|  | 9–11 | 59 | 13.7 |
|  | 12–17 | 153 | 35.6 |
|  | 18–23 | 151 | 35.1 |

## The diet practice for children

Regarding diet practice, 169 (39.3%) children received minimum dietary diversity, and 157 (36.5%) of them received a minimum acceptable diet. Of the total, 135 (39.8%) breast milk-feeding children received a minimum acceptable rating. Regarding non-breastfeeding children, 73 (80%) received MMF and 52 (57%) received MMFF. Among three hundred thirty-nine breastfeeding children, 276 (81.4%) received MMF (Fig 2).

**Table 2. Maternal and child health care-related characteristics of children aged 6–23 months in Dera town, Oromia Ethiopia, March 13 to April 30, 2023. (n = 430).**

| Variables | Category | Frequency | Percent (%) |
|---|---|---|---|
| Frequency of ANC | ≤ 3 | 233 | 54.2 |
| | ≥ 4 | 197 | 45.8 |
| Frequency of PNC | 0 | 130 | 30 |
| | 1–2 | 290 | 67.4 |
| | ≥ 3 | 10 | 2.6 |
| GMP | Yes | 404 | 94 |
| | No | 26 | 6 |
| Place of delivery | Home | 29 | 6.7 |
| | Health facility | 401 | 93.3 |
| Exposure to communication media | Yes | 383 | 89 |
| | No | 47 | 11 |
| Awareness on IYCFP | Yes | 186 | 43.3 |
| | No | 244 | 56.7 |

ANC: Antenatal care; PNC: Postnatal care; GMP: Growth monitoring and promotion; IYCFP: infant and young child feeding practice

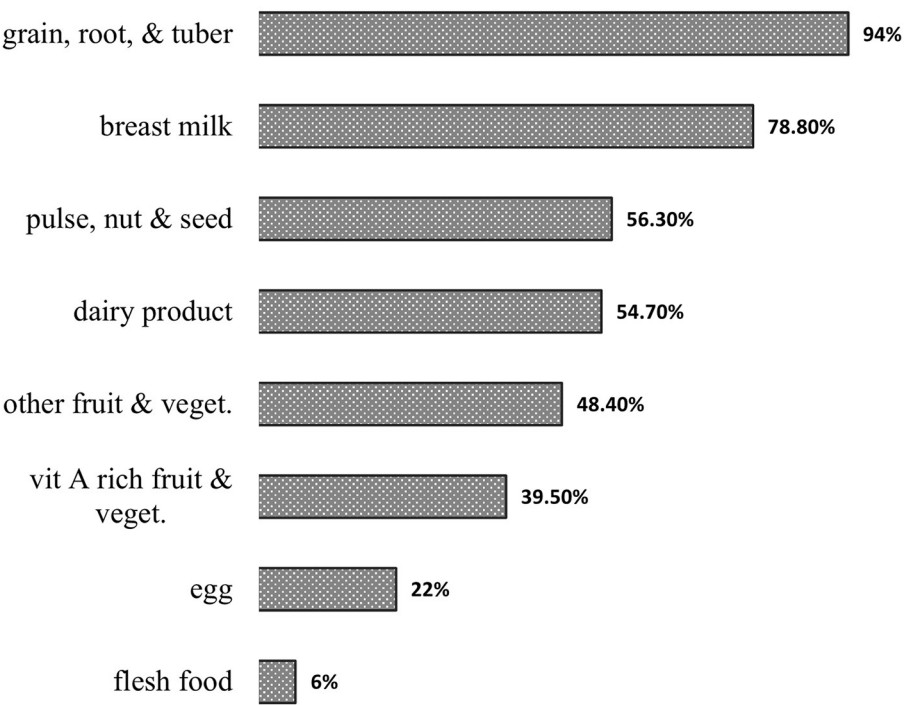

**Fig 1. Consumptions of food groups among children aged 6–23 months in Dera town, Oromia Ethiopia.**

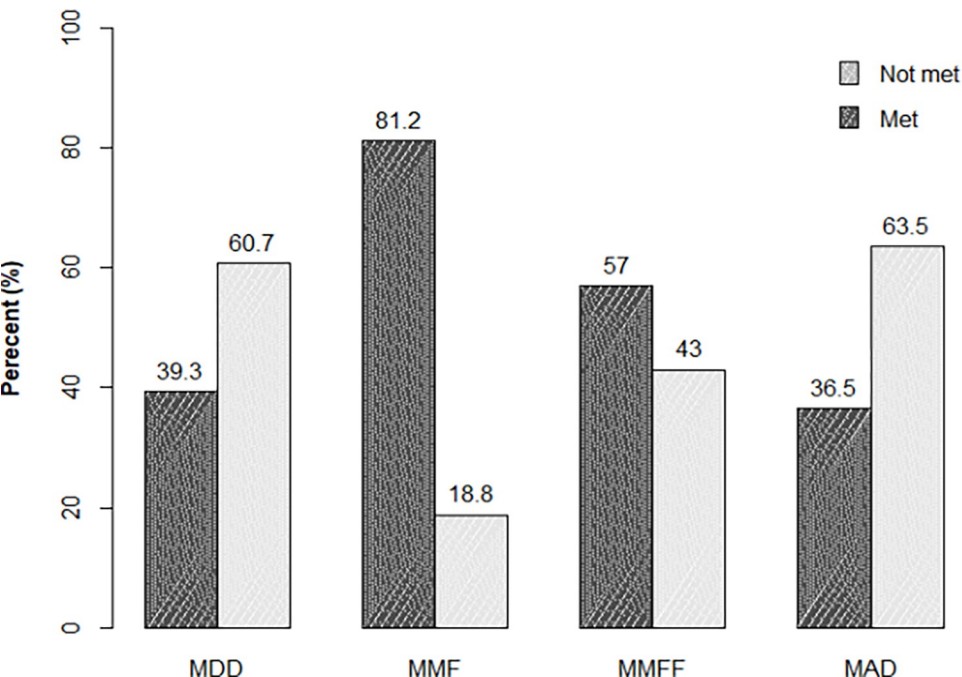

**Fig 2. Distribution of MDD, MMF, MMFF, and MAD among children aged 6–23 months in Dera town, Oromia Ethiopia.**

### Factors associated with the minimum acceptable diet

In the adjusted model, the odds of receiving MAD for IYC aged 6–23 months were 2.67 (AOR = 2.67; 95% CI: 1.25, 5.70) and 4.25 (AOR = 4.25; 95% CI: 1.40, 13.00) times higher among mothers or caregivers who had primary, and college and above-level education, respectively. The odds of gaining a MAD for mothers who had antenatal care visits ≥ 4 were 2 (AOR = 2.0; 95% CI: 1.20, 3.36) times higher than their counterparts. Children aged between 12–17 months (AOR = 6.20; 95% CI: 2.81, 13.50) and between 18–23 months (AOR = 4.61; 95% CI: 2.04, 10.40) had odds of 6.2 and 4.6 times higher in gaining MAD when compared with those aged 6–8 months, respectively. The odds of gaining MAD among non-breast-feeding children were 0.15 (AOR = 0.15; 95% CI: 0.07, 0.31) times less than those who fed breast-milk children (Table 3).

### Discussion

This study was aimed to assess the prevalence and associated factors of the minimum acceptable diet among children aged 6–23 months in Dera town, Oromia, Ethiopia. A preprint has previously been published [Charkos TG_ et.al 2024] [22]. This study revealed that the practice of providing a minimum acceptable diet in our study area was very low 36.5% (95%CI: 32, 41%), according to the World Food Program target. Key predictors of achieving a minimum acceptable diet included the mother's educational status, antenatal care visits, the child's age, and breastfeeding practices.

In our study setting, the practice of providing a minimum acceptable diet for children aged 6–23 months was lower than that reported in a study conducted in Mongolia (43.8%) [23], in rural areas of China (46.4%) [24], and in Latin America and the Caribbean (43.7%) [3]. The possible reasons for the discrepancy might be sample size, study setting, study period, cultural variation related to child feeding practice, and educational and economic differences within

**Table 3. Bivariate and multivariable logistic regression on the minimum acceptable diet practice among children aged 6–23 months, in Dera town, Oromia, Ethiopia, March 13 to April 30, 2023 (n = 430).**

| Variable | Category | Met MAD | Not Met MAD | COR (95%CI) | AOR (95% CI) |
|---|---|---|---|---|---|
| | | N (%) | N (%) | | |
| Ethnicity of mother /caregiver | Oromo | 234(62.4) | 141(37.6) | 1 | 1 |
| | Silte | 20(80) | 5(20) | 0.42(0.15,1.13) | 0.53(0.21,1.97) |
| | Amhara | 19(60) | 11(40) | 0.96(0.44–2.07) | 0.44(0.15,1.24) |
| Religion of mother/caregiver | Muslim | 198(66.4) | 100(33.6) | 1 | 1 |
| | Orthodox | 68(56.7) | 52(43.3) | 1.51(0.65,2.34) | 2.5(0.98,4.8) |
| | Protestant | 7(58.3) | 5(41.7) | 1.40(0.43,4.60) | 1.8(0.4,8.3) |
| Mother/ caregiver education level | Illiterate | 63(80.7) | 15(19.3) | 1 | 1 |
| | Primary | 108(63.5) | 62(36.5) | 2.4(1.26,4.60) | **2.67(1.25,5.70)** * |
| | Secondary | 78(67.8) | 37(32.2) | 2.0(1.00,4.00) | 1.30(0.50,3.01) |
| | ≥ College | 24(35.8) | 43(64.2) | 7.50(3.55,15.97) | **4.25(1.40,13.00)** * |
| Mother/caregiver occupation | Housewife | 221(67.4) | 107(32.6) | 1 | 1 |
| | Merchant | 40(66.7) | 20(33.3) | 1.03(0.57,1.84) | 1.00(0.51,2.01) |
| | Gov't employ | 12(28.6) | 30(62.4) | 5.14(2.53,10.43) | 1.60(0.55,4.77) |
| Father educational status | Illiterate | 32(75) | 11(25) | 1 | 1 |
| | Elementary | 81(68) | 38(32) | 1.36(0.62,2.99) | 0.89(0.35,2.24) |
| | Secondary | 102(71.8) | 40(28.2) | 1.14(0.53,2.50) | 0.57(0.21,1.53) |
| | ≥ College | 58(46) | 68(54) | 3.41(1.60,7.40) | 2.1(0.62,6.85) |
| Occupation of the father | Daily laborer | 80(72.7) | 30(27.3) | 1 | 1 |
| | Farmer | 40(74) | 14(26) | 0.93(0.45,1.95) | 0.77(0.32,1.82) |
| | Merchant | 43(59.7) | 29(40.3) | 1.80(0.96,3.37) | 0.95(0.38,2.40) |
| | Gov't employ | 53(48.2) | 57(51.8) | 2.87(1.64,5.03) | 0.77(0.25,2.37) |
| | private | 57(67.9) | 27(32.1) | 1.26(0.67,2.335) | 1.13(0.50,2.64) |
| Household monthly income | ≤ 1500 | 18(75) | 6(25) | 1 | 1 |
| | 1501–3000 | 63(72.4) | 24(27.6) | 1.14(0.40,3.22) | 0.61(0.18,0.21) |
| | 3001–4500 | 42(69) | 19(31) | 1.36(0.46,3.96) | 0.88(0.24,3.14) |
| | ≥4501 | 150(58) | 108(42) | 2.16(0.83,5.62) | 0.68(0.19,2.45) |
| ANC | ≤ 3 | 168(72.1) | 65(27.9) | 1 | 1 |
| | ≥4 | 105(53.2) | 92(46.8) | 2.30(1.52,3.38) | **2.0(1.20,3.36)** * |
| Age of the child in completed months | 6–8 | 54(80.6) | 13(18.8) | 1 | 1 |
| | 9–11 | 47(79.7) | 12(20.3) | 1.1(0.44,2.55) | 1.4(0.54–3.8) |
| | 12–17 | 80(53.4) | 73(46.4) | 3.79(1.9,7.5) | **6.20(2.81,13.50)** *** |
| | 18–23 | 92(61) | 59(39) | 2.7(1.34,5.30) | **4.61(2.04,10.40)** *** |
| Exposure to communication media | Satisfy (yes) | 237(61.8) | 146(38.2) | 1 | 1 |
| | Unsatisfied (no) | 36(77) | 11(23) | 0.50(0.24,1.00) | 0.62(0.26,1.43) |
| IYCFP education | Yes | 108(69) | 78(31) | 1 | 1 |
| | No | 165(67.6) | 79(32.4) | 0.68(0.46,1.01) | 1.01(0.62,1.67) |
| Breastfeeding status | Yes | 204(60) | 135(40) | 1 | **1** |
| | No | 69(76) | 22(24) | 0.48(0.28,0.82) | **0.15(0.07,0.31)** *** |

MAD: Minimum acceptable dietary; ANC: Antenatal care; IYCFP: IYCFP, infants and young children feeding practice; COR: Crude odds ratio; AOR: adjusted odds ratio; * statistically significant at p-value < 0.05 and *** at p-value < 0.0001 in multivariable logistic regression. 1 = reference category.

the country. Our studies were conducted in areas where culturally similar child feeding was practiced and in less educated communities.

However, this finding is higher than studies conducted in South Asia (12.1%) [3], Philippines (6.7%) [25], Sub-Saharan Africa (10.1%) [17], East Africa (11.56%) [26], Ethiopia (11%),

and Oromia regional state 16.4% [6], Sidama region 13.8% [13], Goncha district, northwest Ethiopia (12.6%) [16] and Sheno town, North Showa, and Oromia (13.6%) [9]. The possible explanation for the difference might be the study period, study setting, sociodemographic characteristics of the study population, geographic variation, cultural beliefs of the countries, economic level of the study area, and the inclusion of breast milk into food groups to assess the dietary diversity of the children aged 6–23 months. The current study was conducted in economically better areas than the previous studies conducted in Ethiopia [6], as a reason a high prevalence of MAD was found. This study was conducted among urban children, a culturally similar population, and included breast milk as one food group according to the revised indicators of the IYCF assessment. In other sides, our findings was lower than studies conducted in Addis Ababa 74.6% [27]. This might be because of the educational level of the mothers, health accessability and coverage, and economic difference. Since, it is conducted in the capital city of Ethiopia.

In the adjusted model, having four or more ANC visits was more likely to provide the recommended minimum acceptable diet. This finding is similar to a study done in sub-Saharan Africa and Debre Berhan town [15, 17]. This is because those mothers who had antenatal care follow-ups might have better nutritional advice and counseling provided by health workers during antenatal care visits.

The study suggested that the educational level of the mothers was significantly associated with a minimally acceptable diet. This finding is similar to a study done in Nepal [28], sub-Saharan Africa [17], Mareka district southern region [29], Aleta Wondo district, Sidama region [13], and Debre Berhan town, central Ethiopia [15]. This might be because mothers may perceive the young child as having the poor ability of the intestine to digest certain foods and perceive that if they get animal milk in addition, breast milk is enough for them during the infancy period, which shows a poorly diversified diet. In this study, having a high level of education was positively associated with a minimum acceptable diet. The findings were in line with the studies done in sub-Saharan Africa [17], east Africa [26] and Mareka district [29], Ethiopian demographic and health survey 2019 [4], Debre Berhan town [15], and Goncha district [16]. This might be because educated mothers were more likely to have information and a better understanding of messages and any information about child-feeding practice and its advantages.

In this study, children who currently fed breast milk were 85% more likely to get a minimum acceptable diet than those who did not feed breast milk. This finding is similar to the study done in Nepal and sub-Saharan Africa [17, 28]. This might be because breast milk is easily and equally available to those who feed breastmilk and need no expense. On the other hand, non-breast milk-fed children should get the minimum amount of milk at least two times per day, which needs extra money and might be impossible for some parents due to economic shortages.

The possible limitations of this study could be that the 24-hour dietary recall method might not show the usual intake and feeding pattern of the child. It is a cross-sectional study; it does not show a cause-and-effect relationship. Since we used a single-day 24-hour food recall questionnaire, we are not able to assess a day-to-day variation of dietary habits. Lastly, the study was conducted in the town, the results may be overestimated to represent the general population of Ethiopia.

## Conclusion

The study showed that the practice of a minimum acceptable diet among children aged between 6–23 months in the study area was low when compared to the World Food Program

target. Having antenatal care follow-up visits more than or equal to four, maternal education level of primary and college and above, child age between 12–17 months, and 18–23 months, not feeding breast milk were significantly associated with the minimum acceptable diet.

## Supporting information

**S1 Fig. Schematic representation for sampling procedure among children aged 6–23 months in Dera town, Oromia Ethiopia.**
(PNG)

**S1 Data.**
(SAV)

## Acknowledgments

The authors would also like to extend their appreciation to all the mothers/caregivers, data collectors, and supervisors.

## Author Contributions

**Conceptualization:** Girma Cheru Bikila, Tesfaye Getachew Charkos.

**Data curation:** Girma Cheru Bikila, Tesfaye Getachew Charkos.

**Formal analysis:** Girma Cheru Bikila, Tesfaye Getachew Charkos.

**Funding acquisition:** Girma Cheru Bikila.

**Investigation:** Girma Cheru Bikila, Tesfaye Getachew Charkos.

**Methodology:** Girma Cheru Bikila, Tesfaye Getachew Charkos.

**Project administration:** Girma Cheru Bikila, Godana Arero, Sultan Kalu, Tesfaye Getachew Charkos.

**Resources:** Girma Cheru Bikila, Tesfaye Getachew Charkos.

**Software:** Girma Cheru Bikila, Tesfaye Getachew Charkos.

**Supervision:** Girma Cheru Bikila, Godana Arero, Sultan Kalu, Tesfaye Getachew Charkos.

**Validation:** Girma Cheru Bikila, Sultan Kalu, Kedir Teji Roba, Tesfaye Getachew Charkos.

**Visualization:** Girma Cheru Bikila, Godana Arero, Sultan Kalu, Kedir Teji Roba, Tesfaye Getachew Charkos.

**Writing – original draft:** Girma Cheru Bikila, Godana Arero, Sultan Kalu, Tesfaye Getachew Charkos.

**Writing – review & editing:** Girma Cheru Bikila, Godana Arero, Sultan Kalu, Kedir Teji Roba, Tesfaye Getachew Charkos.

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
