## [Decision Letter · Decision Letter 0]

5 Jul 2024

PONE-D-24-11210The drivers of minimum acceptable diet among children aged 6–23 months in Ethiopia: community-based cross-sectional studyPLOS ONE

Dear Dr. Charkos,

Thank you for submitting your manuscript to PLOS ONE. After careful consideration, we feel that it has merit but does not fully meet PLOS ONE’s publication criteria as it currently stands. Therefore, we invite you to submit a revised version of the manuscript that addresses the points raised during the review process.

Please submit your revised manuscript by Aug 19 2024 11:59PM. If you will need more time than this to complete your revisions, please reply to this message or contact the journal office at plosone@plos.org. Please include the following items when submitting your revised manuscript:A rebuttal letter that responds to each point raised by the academic editor and reviewer(s). You should upload this letter as a separate file labeled 'Response to Reviewers'.A marked-up copy of your manuscript that highlights changes made to the original version. You should upload this as a separate file labeled 'Revised Manuscript with Track Changes'.An unmarked version of your revised paper without tracked changes. You should upload this as a separate file labeled 'Manuscript'.

We look forward to receiving your revised manuscript.

Kind regards,

Werku Etafa

Academic Editor

PLOS ONE

Journal Requirements:

3. We note that [Figure 1] in your submission contain [map/satellite] images which may be copyrighted. All PLOS content is published under the Creative Commons Attribution License (CC BY 4.0), which means that the manuscript, images, and Supporting Information files will be freely available online, and any third party is permitted to access, download, copy, distribute, and use these materials in any way, even commercially, with proper attribution. For these reasons, we cannot publish previously copyrighted maps or satellite images created using proprietary data, such as Google software (Google Maps, Street View, and Earth). For more information, see our copyright guidelines: http://journals.plos.org/plosone/s/licenses-and-copyright.

Reviewers' comments:

Reviewer's Responses to Questions

**Comments to the Author**

1. Is the manuscript technically sound, and do the data support the conclusions?

Reviewer #1: Yes

2. Has the statistical analysis been performed appropriately and rigorously? 

Reviewer #1: Yes

3. Have the authors made all data underlying the findings in their manuscript fully available?

Reviewer #1: Yes

4. Is the manuscript presented in an intelligible fashion and written in standard English?

Reviewer #1: Yes

5. Review Comments to the Author

Reviewer #1: Comment for Author

- The written topic and objectives are not consensus for example you need to assess the drives factors or prevalence which one is your action verb?

- You have to spell out the abbreviation in the first text e.g. EMDHS

- Under introduction the fact about the topic is not well intervened, for example what about MDDS from global to local has to be well explained,

- Better to focus on the objective of the study and the knowledge you have generated has to be also around you research question and gap, therefore it has to be focused and narrated as well as sequenced

- What are factor contributing for your study topic? Previous experience to alleviate the problem? Things to be clear/researched?

- Why this problem is a problem?

Methods:

- The study area crop production top graph etc. has to be mentioned

- Why you took a margin of error 3%? What it effect on sample size determination? Do you have any reference or justification which is scientifically sound?

- Sampling procedure is not clear, it need detail description, how you employed this approach and how the study populations are selected including schematic presentation and sampling frame.

- How food frequency question measure MDDs? What is the difference between FFQ and 24hr food recall methods? Which one is best? How? Can we use them simultaneously? How?

- How many day 24hr food recall were recruited? What is the impact of single 24hr food recall on the dietary diversity measurement compared to other additional day?

- What are methods used to control measurement error on both approach you have used? It has to be explained well

- Where is the eight food group you have developed or adopted? With reference

- What action or procedure you did to reduce measurement error?

- What do you mean local language? Or you mean that largely spoken language in the area?

- What is MAD? Its difference with MDDS? Uniformity is must throughout the document

Result

- You have report the age of the mother by median, what is difference between mean and median? why report by median?

- What is your reference for “Household monthly income” classification?

- Do you thing that the word majority explains “The majority of the mothers (89%)” this value? Look at such error and in appropriate use of words here and there in your document

- How the “minimum meal frequency” is calculated for different age group and computed for MAD?

- “Exposure to communication media”, what do you mean? What are the types of media? Which one is yes which one is no?

- The model fitness and multicollinarity issue was not explained

Discussion

- The first paragraph should be the summary of you finding and objectives

- The logical explanation should be scientifically sound or it has to had a reference

6. PLOS authors have the option to publish the peer review history of their article (what does this mean?). If published, this will include your full peer review and any attached files.

Reviewer #1: No

---

## [Author Response · Author response to Decision Letter 0]

2 Aug 2024

Response to Reviewers

PONE-D-24-11210

The prevalence and associated factors of the minimum acceptable diet among children aged 6–23 months in Ethiopia: a community-based cross-sectional study

Thank you for submitting your manuscript to PLOS ONE. After careful consideration, we feel that it has merit but does not fully meet PLOS ONE’s publication criteria as it currently stands. Therefore, we invite you to submit a revised version of the manuscript that addresses the points raised during the review process.

Response: Thank you for your positive comments and for showing interest in our manuscript.

Response: As suggested, we have removed the study area map from our manuscript. While the map provided a visual representation of the study area, its exclusion does not impact our results. 

Reviewer #1: Comment for Author

“The written topic and objectives are not consensus for example you need to assess the drives factors or prevalence which one is your action verb?”

Response: Thank you for pointing this out. The objective of our study was to determine the driving factors and prevalence of the minimum acceptable diet among children aged 6–23 months. Following the reviewer's suggestion, we have revised the title to: “The prevalence and associated factors of the minimum acceptable diet among children aged 6–23 months in Ethiopia: a community-based cross-sectional study.”

“You have to spell out the abbreviation in the first text e.g. EMDHS”

Response: Thanks for your notice. It has now been revised as: “According to the 2019 Ethiopia mini demographic and health survey (EMDHS). (page #2)

“Under introduction the fact about the topic is not well intervened, for example what about MDDS from global to local has to be well explained, Better to focus on the objective of the study and the knowledge you have generated has to be also around you research question and gap, therefore it has to be focused and narrated as well as sequenced”

Response: Thank you for the reviewer's concern. However, our aim in this study is not to focus solely on minimum dietary diversity score (MDDS) among children, but rather on the combination of minimum meal frequency (MMF) and minimum dietary diversity (MDD), which together constitute the minimum acceptable diet (MAD). Our primary objective is to elucidate the prevalence and associated factors of the MAD in Ethiopia, which we have detailed in the introduction. As stated in the second paragraph of the introduction, we provide an overview of the MAD problem from a global to a local perspective. “Globally, only two-thirds of 6-8-month-olds are receiving any solid food at all; of this, 1 in 2 children receives a minimum meal frequency, and less than 1 in 3 receives minimum dietary diversity. Considering both minimum meal frequency and minimum diet diversity, only about 1 in 6 children receives a minimally acceptable diet [1, 2]. In developing countries, 1 in 5 children aged 6 to 23 months is fed the minimum recommended /acceptable diverse diet [3]. According to the 2019 Ethiopia mini demographic and health survey (EMDHS), 11 % of children aged 6-23 months meet the minimum standards concerning all three infant and young child feeding (IYCF) practices (breastfeeding status, number of food groups, and times) they fed during the day or night before the survey. This indicates only 1 in 9 children receives a minimally acceptable diet, which is very low compared to the national recommendation [4].”

“What are factor contributing for your study topic? Previous experience to alleviate the problem? Things to be clear/researched? “ 

Response: Several contributing factors were identified during our literature review of previously published studies. All variables included in our analysis were considered as a possible potential factors for our study topic. Despite the Ethiopian government's policy and strategy efforts to improve the minimum acceptable diet for children, the issue remains unresolved. Various reasons have been suggested for this ongoing issue, including political unrest, food insecurity, unplanned family size, and drought.

“Why this problem is a problem? “

Response: As suggested by the reviewers, we have incorporated a paragraph addressing the main problem of minimum acceptable diet (MAD). It is readable as “In developing countries like Ethiopia, high levels of poverty contribute to persistent food insecurity, leading to various health problems. Inadequate feeding practices during infancy and childhood can result in delayed growth and development, stunting, wasting, and undernutrition, which, in turn, increase morbidity and mortality among children under five. Therefore, it is crucial to assess the prevalence and contributing factors of MAD among children aged 6-23 months to mitigate the adverse outcomes associated with inadequate nutritional practices.” (page #3)

Methods:

“ The study area crop production top graph etc. has to be mentioned”

Response: As suggested, we have now included a detailed discussion on the agricultural production in the study area. That is readble as “The district is located in the Great Rift Valley and has an altitude of 1400-2500 meters above sea level with a climatic condition of 99% desert and 1% temperate. Mostly cultivated grains are wheat, teff, maize, and barley, and mostly cultivated vegetables are onions and garlic.” (page #3)

“Why you took a margin of error 3%? What it effect on sample size determination? Do you have any reference or justification which is scientifically sound?

Response: Thanks for your concern. The sample size determination for a single population proportion formula relies on the previous prevalence of cases. Optimal sample size is achieved when the previous prevalence (p) of the case is close to 0.5 [5]. When the prevalence of the case (P) is significantly less or greater than 0.5, the maximum sample size is not achieved. In our study, the prevalence of the MAD was 13.8% [6], based on data from Aleta Wondo District in the Sidama Region, which is considerably lower than 0.5. Consequently, using the standard margin of error of 0.05 (5% level of significance), the calculated sample size would be smaller. To address this issue and to increase the sample size, we opted to decrease the level of significance (margin of error) from 5% to 3%. With this adjustment, our sample size increased from 182 to 436, in which our final smaple size was 436 subjects. 

“Sampling procedure is not clear, it need detail description, how you employed this approach and how the study populations are selected including schematic presentation and sampling frame.”

Response: As suggested by the reviewers, we have now expressed in details the sampling procedure in the manuscript as follows: “Dera town comprises two kebeles, both of which were selected purposively for this study. Study participants were chosen using systematic random sampling. The number of children aged 6 to 23 months with mothers/caregivers during the study period was obtained from the Dera Health Center EPI registration. The sample size was proportionally allocated based on the number of eligible children in each kebele.

The first respondent was randomly selected using the lottery method. Subsequent respondents were selected every Kth interval, where K = N/n = 1804/436 = 4 (N being the total population in the study area and n being the sample size). From each household, one eligible child with a mother/caregiver was selected. If more than one eligible child was present, one was randomly chosen by lottery method. This process was repeated until the required sample size for both kebeles was reached. If a mother/caregiver was not available on the date of data collection, the next mother was approached after two subsequent visits (S1 Fig). (page # 5)

 “How food frequency question measure MDDs?” 

Response: A food frequency questionnaire (FFQ) measures a minimum dietary diversity (MDD) by assessing how often specific food items or food groups are consumed over a specified period, typically the past week or month. Respondents indicate the frequency of consumption for various food items, which can then be categorized into predefined food groups necessary for evaluating dietary diversity. The MDD for children aged 6-23 months is typically defined as consuming foods from at least five out of eight standard food groups within a given time period.

“What is the difference between FFQ and 24hr food recall methods?”

Response: A FFQ assesses usual dietary intake over a longer period, such as a week, month, or year (retrospective recall), and the respondents report the frequency of consumption for a list of food items or food groups. It is less detailed about specific quantities or exact intake on a given day.

While a 24-hour food recall was used to assesses all food and beverages consumed in the past 24 hours, and it is a very detailed about specific items and quantities.

“Which one is best? How?”

Response: The best methods were depends on the researcher objectives. When the researcher aimed to determine the long-term dietary patterns (chronic dietary) behaviors and their relationship to health outcomes the food frequency questionnaire is the best method. While, if the researcher aimed to assess the dietary habit on a specific day, the 24-hour recall method was the best method. 

“Can we use them simultaneously? How?”

Response: Yes, using both FFQ and 24-hour recall simultaneously can be beneficial to leverage the strengths of each method and compensate for their weaknesses. 

“How many day 24hr food recall were recruited? What is the impact of single 24hr food recall on the dietary diversity measurement compared to other additional day?”

Response: The number of days for 24-hour food recall can vary depending on the research resources. In this study, due to financial constraints, we collected only a single day of 24-hour food recall data from the participants. Of course, relying on a single day's recall significantly impacts the precision of the information obtained, as it cannot capture day-to-day variations. This approach may lead to underestimation or overestimation of dietary diversity depending on the foods consumed on that specific day. As a result, we have now included this in the limitation parts. It is readable as “Since we used a single-day 24-hour food recall questionnaire, we are not able to assess a day-to-day variations of dietary habits.” (page # 14) 

“What are methods used to control measurement error on both approach you have used? It has to be explained well”

Response: Thanks for your concern. To minimize measurement error during data collection, we implemented rigorous procedures: we utilized a standardized questionnaire adapted from WHO guidelines, trained data collectors thoroughly, pretested the questionnaire prior to data collection, translated the questionnaire into Afan Oromo (the participants' language), and ensured meticulous supervision to verify data completeness. These steps were designed to enhance the accuracy and reliability of our dietary assessment process. This was already mentioned in dteails in the manuscript. 

“Where is the eight food group you have developed or adopted? With reference”

Response: Thanks for your notice. We have already provided a clear list of the eight food groups included in the operational definition, along with their respective references. “1. Breast milk, 2. Grains; roots; tubers and plantains, 3. Pulses (beans, peas, lentils); nuts and seeds, 4. Dairy products (milk, infant formula, yogurt, cheese), 5. Flesh foods (meat, fish, poultry, organ meats), 6. Eggs, 7. Vitamin-A-rich fruits and vegetables and 8. Other fruits and vegetables. [7]

 “What action or procedure you did to reduce measurement error?”

Response: Thanks for your concern. To minimize measurement error during data collection, we implemented rigorous procedures: we utilized a standardized questionnaire adapted from WHO guidelines, trained data collectors thoroughly, pretested the questionnaire prior to data collection, translated the questionnaire into Afan Oromo (the participants' language), and ensured meticulous supervision to verify data completeness. These steps were designed to enhance the accuracy and reliability of our dietary assessment process.

‘What do you mean local language? Or you mean that largely spoken language in the area?”

Response: Thanks for your concern, we mean that local language Afan Oromo (the participants' language). Now we have revised and readable as “The questionnaire was translated into the local language (the participants' language) Afan Oromo by a language expert and translated into English to validate its consistency.) (page # 5)

- What is MAD? Its difference with MDDS? Uniformity is must throughout the document

Response: MAD is an indicator used to assess the quality of a child's diet. It considers both the dietary diversity and the feeding frequency appropriate for the child's age. Specifically, MAD measures whether children aged 6-23 months receive a MAD that meets both the minimum dietary diversity (MDD) and minimum meal frequency (MMF) criteria.

While, 

MDD is a measure of dietary diversity alone. It indicates whether children aged 6-23 months have consumed foods from at least five out of eight specific food groups within a given period. On the other hand, it does not concider frequency, only measure diversity of the food taken. Therefore, it is quit differ with MAD. MAD considers diversity and frequency of food taken. 

 Of course, this study primarily focuses on the minimum acceptable diet (MAD) among children aged 6-23 months. Consequently, the manuscript thoroughly explains and discusses MAD from the introduction through to the discussion section.

Result

- You have report the age of the mother by median, what is difference between mean and median? why report by median?

Response: We thanks for reviewer’s concerns. We used the median for the mothers' ages because the data was skewed and did not meet the normality assumption according to the Shapiro-Wilk test. Therefore, the median is the most appropriate measure for skewed data.

“What is your reference for “Household monthly income” classification?”

Response: Thanks for your comments. We classified household monthly income based on a previous study conducted in Ethiopia. Given that Ethiopia is one of the least developed countries, government employee salaries are significantly lower compared to other countries. Additionally, the characteristics of our study area influenced this classification method.

- Do you thing that the word majority explains “The majority of the mothers (89%)” this value? Look at such error and in appropriate use of words here and there in your document

Response: As suggested, now we have revised and readable as “Of the total participants, 383 (89%), mothers had exposure to communication media, while 43.3% had awareness about infant and young child feeding practice (IYCFP)” (page # 9) 

“How the “minimum meal frequency” is calculated for different age group and computed for MAD?”

Response: We have operationalized the minimum meal frequency and its calculation as follows “Minimum meal frequency: percentage of children 6–23 months of age who consumed solid, semi-solid, or soft foods (but also including milk feeds for non-breastfed children) at least the minimum number of times during the previous day [4, 7, 8]. 

The minimum number of times is defined as:

Two feedings of solid, semi-solid, or soft foods for breastfeeding infants aged 6–8 months and three feedings of solid, semi-solid, or soft foods for breastfeeding children aged 9–23 months. Four feedings of solid, semi-solid, or soft foods or milk feeds for non-breastfeeding children aged 6–23 months whereby at least one of the four feeds must be a solid, semi-solid, or soft feed [7-9].” (page # 7)

 “Exposure to communication media”, what do you mean? What are the types of media? Which one is yes which one is no?

Response: Thanks for your notice. Communication media encompasses a wide range of channels and tools used to convey information related to child feeding practices, such as digital media, social media, print media, broadcast media, and mobile media. In our study, if mothers had exposure to at least one of these media, t

---

## [Decision Letter · Decision Letter 1]

21 Nov 2024

The prevalence and associated factors of the minimum acceptable diet among children aged 6–23 months in Ethiopia: community-based cross-sectional study

PONE-D-24-11210R1

Dear Dr. Tesfaye Getachew Charkos,

We’re pleased to inform you that your manuscript has been judged scientifically suitable for publication and will be formally accepted for publication once it meets all outstanding technical requirements.

Kind regards,

Yibeltal Alemu Bekele, MpH

Academic Editor

PLOS ONE

Additional Editor Comments (optional):

Reviewers' comments:

Reviewer's Responses to Questions

**Comments to the Author**

1. If the authors have adequately addressed your comments raised in a previous round of review and you feel that this manuscript is now acceptable for publication, you may indicate that here to bypass the “Comments to the Author” section, enter your conflict of interest statement in the “Confidential to Editor” section, and submit your "Accept" recommendation.

Reviewer #1: All comments have been addressed

2. Is the manuscript technically sound, and do the data support the conclusions?

Reviewer #1: (No Response)

3. Has the statistical analysis been performed appropriately and rigorously? 

Reviewer #1: (No Response)

4. Have the authors made all data underlying the findings in their manuscript fully available?

Reviewer #1: (No Response)

5. Is the manuscript presented in an intelligible fashion and written in standard English?

Reviewer #1: (No Response)

6. Review Comments to the Author

Reviewer #1: Than k you very much for the chance, I have seen the responded comment by the author through out the article, he has responded all concern i have on the article and i accepted it to be published on PLOS ONE journal, he did as per the criteria and guideline set by the journal editorial standard set. All the best

7. PLOS authors have the option to publish the peer review history of their article (what does this mean?). If published, this will include your full peer review and any attached files.

Reviewer #1: No

---

## [Editor Report · Acceptance letter]

12 Dec 2024

PONE-D-24-11210R1 

PLOS ONE

Dear Dr. Charkos, 

I'm pleased to inform you that your manuscript has been deemed suitable for publication in PLOS ONE. Congratulations! Your manuscript is now being handed over to our production team.

Kind regards, 

on behalf of

Mr. Yibeltal Alemu Bekele 

Academic Editor

PLOS ONE